# GeoVideo: Introducing Geometric Regularization into Video Generation Model

**Yunpeng Bai**[1]    **Shaoheng Fang**[1]    **Chaohui Yu**[2,3]    **Fan Wang**[2]    **Qixing Huang**[1]

[1]The University of Texas at Austin, [2]DAMO Academy, Alibaba Group, [3]Hupan Lab
`https://geovideo.github.io/GeoVideo/`

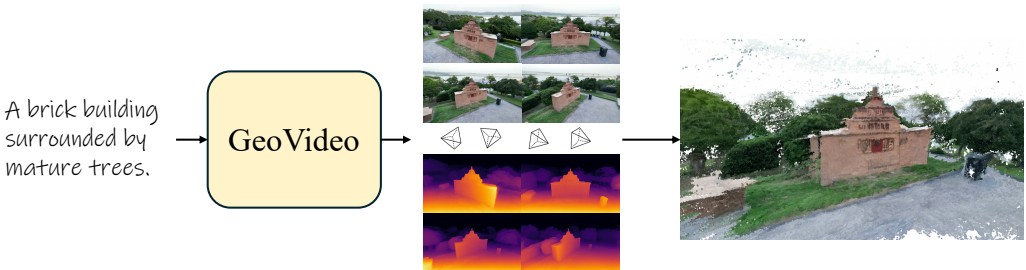

Figure 1: GeoVideo introduces a geometric consistency loss using predicted depths and camera poses to enhance multi-view consistency of the output frames, leading a high-quality 3D reconstruction from the output video frames.

## Abstract

Recent advances in video generation have enabled the synthesis of high-quality and visually realistic clips using diffusion transformer models. However, most existing approaches operate purely in the 2D pixel space and lack explicit mechanisms for modeling 3D structures, often resulting in temporally inconsistent geometries, implausible motions, and structural artifacts. In this work, we introduce geometric regularization losses into video generation by augmenting latent diffusion models with per-frame depth prediction. We adopted depth as the geometric representation because of the great progress in depth prediction and its compatibility with image-based latent encoders. Specifically, to enforce structural consistency over time, we propose a multi-view geometric loss that aligns the predicted depth maps across frames within a shared 3D coordinate system. Our method bridges the gap between appearance generation and 3D structure modeling, leading to improved spatio-temporal coherence, shape consistency, and physical plausibility. Experiments across multiple datasets show that our approach produces significantly more stable and geometrically consistent results than existing baselines.

## 1   Introduction

Video generation has recently made significant strides in creating visually impressive and high-quality clips [4, 28, 69, 29, 53, 77]. Powered by diffusion transformer models [39] and large-scale training datasets [2, 63, 8, 61], these systems are capable of synthesizing realistic videos conditioned on various inputs, such as text prompts [26, 35, 26, 62] or images [12, 43, 57, 38]. Despite these successes, current video generation models often fail to accurately capture the underlying geometry, coherent motions, and physical consistency in dynamic scenes. As a result, the generated videos,

39th Conference on Neural Information Processing Systems (NeurIPS 2025).

although plausible at first glance, often exhibit temporal artifacts such as shape deformation, structure flickering, and implausible object interactions.

This limitation stems from the fact that most video generation models operate purely in the 2D pixel/latent space and rely on temporal attention to promote cross-frame coherence. Although effective in maintaining short-term consistency, these approaches lack explicit mechanisms to model 3D structures, leading to violations of object permanence, shape integrity, and motion realism.

This shortcoming highlights a deeper insight: realistic video generation demands more than visual coherence—it requires a structured understanding of the 3D world. After all, videos can naturally encode spatio-temporal observations of real environments. Viewed through this lens, video generation can be reframed as a form of *world modeling*—the construction of continuous, physically grounded representations of the dynamic world. Emerging research on world generation [4, 5] underscores its potential in applications such as 3D scene synthesis [34, 70, 44], robotics [64, 20, 56], and embodied AI [41, 45, 14]. However, achieving physically plausible world modeling requires the generated scenes to maintain consistent geometry over time, which current models struggle to enforce.

To address these limitations, we propose **GeoVideo**, which introduces *geometric regularization* into the video generation process. Specifically, we augment the generative model to predict per-frame depth maps alongside RGB frames and enforce cross-frame depth consistency. This regularization encourages the model to maintain coherent 3D geometry throughout the video. By aligning predicted depth across consecutive frames, the model is guided to better capture the underlying scene structure, resulting in enhanced realism, temporal stability, and physical plausibility. Our key insight is that depth consistency offers implicit geometric supervision that complements appearance-based learning. This helps bridge the gap between 2D frame-level synthesis and 3D-consistent scene modeling, paving the way toward more structured and physically grounded video generation.

The main contributions of this work are:

- **Explicit Geometry Modeling in Video Generation.** We introduce per-frame depth prediction into latent diffusion-based video generation models, enabling explicit modeling of 3D scene structure throughout the generation process.
- **Geometric Regularization.** We propose a cross-frame consistency loss that lifts predicted depths into a global 3D point cloud and supervises them via multi-view reprojection alignment, encouraging globally coherent geometry.
- **Improved Spatiotemporal Coherence.** Our approach significantly enhances structural consistency, motion stability, and geometric plausibility in generated videos, as demonstrated on several benchmarks.

## 2 Related Work

### 2.1 Diffusion Models for Video Generation

The remarkable success of diffusion models in image generation [42, 47, 46] has recently inspired their extension to video generation [25, 65, 13], where they have quickly become the dominant paradigm. In particular, latent diffusion [55, 46] has emerged as a widely adopted strategy: a VAE [27] module first encodes video data into a compact latent space, and the diffusion process is performed within this lower-dimensional representation. Current state-of-the-art methods [69, 29, 76] utilize a 3D Variational Autoencoder [69] in combination with a Diffusion Transformer (DiT) [39] backbone, achieving highly realistic and high-fidelity video synthesis. Despite these advances, existing diffusion-based video generation models [18, 3] often struggle to accurately capture geometric structures, coherent temporal motions, and physical consistency. To mitigate these limitations, recent research has explored the introduction of additional priors to guide the generation process toward better alignments with real-world dynamics.

For example, Track4Gen [22] jointly models video generation and point tracking across frames within a single network, providing enhanced spatial supervision over diffusion features and improving both motion and structural consistency. Similarly, VideoJAM [6] learns a joint appearance-motion representation that instills an effective motion prior to the video generator. Other contemporary approaches have also taken advantage of motion representations to improve motion coherence in image-to-video generation tasks [50, 59]. Meanwhile, OmniVDiff [66] models appearance, depth,

Canny edges, and semantic segmentation simultaneously, enabling multi-modal video generation and multi-modal conditional generation. However, it does not explicitly impose priors on the generated auxiliary signals. IDOL [73] proposes a human-centered joint video-depth generation framework, but it does not introduce explicit priors and is limited to human-centered scenarios. WVD [75] supports the simultaneous generation of appearance and point representations but is restricted to image-to-video translation in small static environments. Complementary to these efforts, Yue et al. [72] proposed to lift the semantic characteristics of each frame into a 3D Gaussian representation, demonstrating that fine-tuning a foundation model with these 3D-aware characteristics leads to better performance across downstream tasks. Building on these insights, our work seeks to jointly model geometry during video generation and introduce an explicit geometric regularization loss to further improve the quality, consistency, and realism of synthesized videos.

## 2.2 Geometry-Related Tasks with Pretrained Diffusion Models

Another line of work uses pre-trained generative models [46, 3, 67] for geometry-related tasks. A pioneering effort in this direction is Marigold [24], which first proposed treating depth as an image-like modality. By encoding depth maps into the same latent space as RGB images using a latent diffusion VAE, Marigold demonstrated that image generation models can be repurposed into depth estimators. This idea has inspired a series of subsequent works [10, 1] that exploit the strong priors of diffusion models to estimate geometric properties such as depth and surface normals. Following this direction, DepthCrafter [19] and Depth Any Video [68] adapt video generation models to perform video-based depth estimation, effectively extending Marigold's latent-space strategy from images to videos. Similarly, DiffusionRender [32] leverages video generation models for inverse rendering tasks, jointly recovering not only scene geometry but also materials and lighting from video sequences. Building on these advancements, Geo4D [23] further expands the use of video generators to tackle 4D scene reconstruction, capturing dynamic scene structures over time.

## 2.3 3D Scene Generation

The generation of 3D content has also emerged as a highly active area of research. The early text-to-3D scene generation methods [17, 9] mainly relied on image generation inpainting models and progressively completed a scene through multiple iterations, resulting in limited efficiency and quality. Subsequent methods design specialized models for text-to-3D scene generation. Director3D [30] proposes a text-to-3D generation framework capable of synthesizing both real-world 3D scenes and adaptive camera trajectories. Recently, some methods like SplatFlow [11] and Bolt3D [52] have also leveraged intermediate 3D representations to directly generate scenes in the form of 3D Gaussian Splatting, either through multiview diffusion or flow matching models. Prometheus [71] introduces a multiview diffusion model based on an RGB-D latent space to generate 3D Gaussian scenes. However, since Prometheus relies primarily on image-based models, the quality and fidelity of the generated 3D content remain limited. Apart from the methods mentioned above, LDM3D [51] is a previous work similar to ours that re-trains Stable Diffusion, but its scope is limited to the latent space of RGB-D images. Our method incorporates geometric regularization into the video generation model, enabling the direct extraction of high-quality 3D scenes from the generated videos.

# 3 Approach

We begin by reviewing the preliminaries of the video diffusion models in Section 3.1. Section 3.2 and Section 3.3 then describe the proposed RGB-D video generation model and our novel geometric regularization loss, respectively. Finally, Section 3.4 introduces the training procedure.

## 3.1 Preliminaries: Video Diffusion Models

Our method is based on latent diffusion-based video generation models, particularly those with 3D VAE backbones such as CogVideoX [69], and transformer-based denoisers such as DiT [39]. These models encode video frames into a compact latent space and apply denoising diffusion to synthesize temporally coherent video sequences. Formally, let $\mathbf{x}_{1:T}$ denote a video clip of $T$ frames, and let $\mathbf{z} = E(\mathbf{x}_{1:T})$ be its latent representation encoded by a VAE. Latent diffusion models define a forward noising process over $\mathbf{z}$ by gradually adding Gaussian noise: $q(\mathbf{z}^{(t)} \mid \mathbf{z}^{(t-1)}) =$

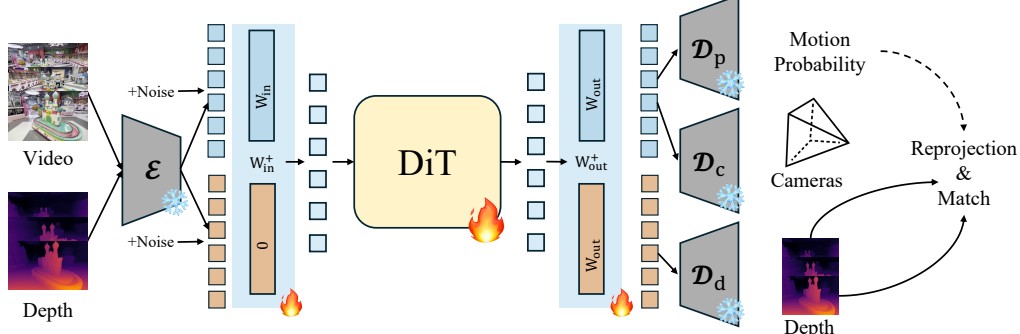

Figure 2: **Overview of the proposed method.** The **orange-yellow** modules are additionally added components. $\mathcal{D}_p$, $\mathcal{D}_c$, and $\mathcal{D}_d$ respectively denote the heads for decoding motion probability, camera pose, and depth from the generated features. We use their outputs to define a geometric consistency loss to regularize the video generative model.

$\mathcal{N}(\mathbf{z}^{(t)}; \sqrt{\alpha_t}\mathbf{z}^{(t-1)}, (1 - \alpha_t)\mathbf{I})$, where $t = 1, \ldots, S$ indexes the diffusion timestep and $\alpha_t$ is a variance schedule. The model learns a reverse process to sample denoised latents: $p_\theta(\mathbf{z}^{(t-1)} \mid \mathbf{z}^{(t)}) = \mathcal{N}(\mathbf{z}^{(t-1)}; \mu_\theta(\mathbf{z}^{(t)}, t), \Sigma_\theta(t))$, which is parameterized by a spatio-temporal transformer (DiT). After obtaining the denoised latent $\mathbf{z}^{(0)}$, the final RGB video is reconstructed via the VAE decoder.

## 3.2 RGBD Video Modeling

To incorporate explicit geometric structure into the video generation process, we choose to represent scene geometry using per-frame depth maps. This choice is motivated by two key factors. First, recent advances [7, 19] in video depth estimation have yielded robust and high-quality predictions, making it feasible to obtain reliable depth supervision even in unconstrained settings. Second, depth maps have a natural image-like structure and can be efficiently encoded into the same latent space as RGB frames using a shared VAE encoder. This design has been validated in prior works such as Marigold [24] and DepthCrafter [19]. Using this modality compatibility, we can extend existing latent video diffusion models to jointly generate RGB and depth with minimal architectural modifications.

As shown in Figure 2, given a pair of RGB and depth frames $(\mathbf{x}_{1:T}^{\text{RGB}}, \mathbf{x}_{1:T}^{\text{D}})$, we encode them in a shared latent space:

$$\mathbf{z} = [\mathbf{z}^{\text{RGB}}; \mathbf{z}^{\text{D}}] = [E(\mathbf{x}_{1:T}^{\text{RGB}}); E(\mathbf{x}_{1:T}^{\text{D}})],$$

where $E(\cdot)$ denotes the VAE encoder and $[\cdot; \cdot]$ denotes channel-wise concatenation. The diffusion model operates on the latent sequence $\mathbf{z}_{1:T}$ and learns to jointly denoise both modalities. The generated latent is then decoded by the VAE decoder into video frames and depth maps.

We model the joint distribution over RGB and depth frames as:

$$P(\mathbf{x}_{1:T}^{\text{RGB}}, \mathbf{x}_{1:T}^{\text{D}}) = P(\mathbf{z}) \prod_{t=1}^{T} P(\mathbf{x}_t^{\text{RGB}}, \mathbf{x}_t^{\text{D}} \mid \mathbf{z}),$$

where each pair is generated from the same underlying latent representation. This formulation enables a tight coupling between appearance and geometry throughout the generation process.

## 3.3 Introducing Geometric Regularization

Although per-frame depth maps provide localized 3D cues, they do not guarantee cross-frame consistency. To enforce coherent 3D structure over time, we introduce a geometric regularization loss that lifts predicted depths into world coordinates using known camera intrinsics and extrinsics. Since depth and appearance are decoded from the same underlying features, applying supervision on the depth prediction allows us to enhance the geometric consistency of the underlying shape without needing to account for view-dependent appearance differences across views.

**Global point cloud construction.** Since the built-in 3D VAE in video generation models is computationally heavy, we additionally train a lightweight decoder $\mathcal{D}_d$ to convert the depth latent $\mathbf{z}^D$ into depth map $\mathbf{D} \in \mathbb{R}^{H \times W}$ of the same resolution as the RGB frame, enabling more fine-grained and accurate geometric regularization. In parallel, and inspired by VGGT [58], we also predict the camera intrinsics and extrinsics for each frame using a camera head from generated $\mathbf{z}^{\text{RGB}}$. Let $\mathbf{D}_i$ be the predicted depth map for frame $i$, and $\mathbf{P}_i \in \text{SE}(3)$ its camera pose. Using the intrinsic matrix $K$, we backproject depth into the camera space and transform it into the world space:

$$\mathbf{X}_i = \mathbf{P}_i \cdot \pi^{-1}(\mathbf{D}_i, K), \tag{1}$$

where $\pi^{-1}$ denotes backprojection from depth to 3D coordinates. The global point cloud is then obtained by aggregating:

$$\mathcal{X}_{\text{global}} = \bigcup_{i=1}^{T} \mathbf{X}_i. \tag{2}$$

We denoise $\mathcal{X}_{\text{global}}$ using voxel grid downsampling and statistical outlier removal to improve robustness and computational efficiency.

**Depth reprojection consistency.** To supervise consistency, we reproject the global point cloud back to each frame and compare its depth with the predicted depth map. For frame $i$, we project:

$$\hat{\mathbf{D}}_i(\mathbf{u}) = \pi_z(\mathbf{P}_i^{-1} \cdot \mathbf{x}), \quad \mathbf{x} \in \mathcal{X}_{\text{global}}, \tag{3}$$

where $\pi_z(\cdot)$ denotes depth value after projection into image coordinates $\mathbf{u}$. We then compute the loss:

$$\mathcal{L}_{\text{geo}} = \frac{1}{T} \sum_{i=1}^{T} \frac{1}{|\mathcal{V}_i|} \sum_{\mathbf{u} \in \mathcal{V}_i} \mathbb{1}(|\hat{\mathbf{D}}_i(\mathbf{u}) - \mathbf{D}_i(\mathbf{u})| < \delta) \cdot |\hat{\mathbf{D}}_i(\mathbf{u}) - \mathbf{D}_i(\mathbf{u})|, \tag{4}$$

where $\mathcal{V}_i$ is the set of valid pixels and $\delta$ is a tolerance threshold set to 0.05. This encourages global shape consistency by penalizing depth discrepancies only when local reprojections are reliable. When multiple points project to the same pixel, we use the average of these points to compute loss. For dynamic videos, we introduce an additional head $\mathcal{D}_p$ that predicts an object movement probability map [31] from the generated video features $\mathbf{z}^{\text{RGB}}$, representing pixels that correspond to dynamic content based on multi-frame information. For dynamic pixels identified in the probability map, we only align them with points of similar probability in adjacent frames.

### 3.4 Parameters Initialization and 2-stage Fine-tuning with Geometric Regularization

**Parameters Initialization.** To enable the pretrained video generation model to support dual-modality inputs (RGB and Depth), we modify the input and output projection layers of the transformer. Let $W_{\text{in}} \in \mathbb{R}^{C_v \times C_t}$ be the input projection matrix, where $C_v$ is the input feature dimension and $C_t$ is the transformer token dimension. Inspired by ControlNet [74], we extend it by vertically concatenating a zero matrix of the same shape, resulting in $W_{\text{in}}^+ \in \mathbb{R}^{2C_v \times C_t}$. The associated input bias $b_{\text{in}} \in \mathbb{R}^{C_t}$ is left unchanged. On the output side, let $W_{\text{out}} \in \mathbb{R}^{C_t \times C_v}$ denote the output projection matrix. To accommodate dual outputs, we horizontally concatenate a copy of $W_{\text{out}}$, resulting in $W_{\text{out}}^+ \in \mathbb{R}^{C_t \times 2C_v}$. The output bias $b_{\text{out}} \in \mathbb{R}^{C_v}$ is duplicated to form $b_{\text{out}}^+ \in \mathbb{R}^{2C_v}$. The initialization is summarized as:

$$W_{\text{in}}^+ = \begin{bmatrix} W_{\text{in}} \\ \mathbf{0} \end{bmatrix} \in \mathbb{R}^{2C_v \times C_t}, \quad b_{\text{in}}^+ = b_{\text{in}} \in \mathbb{R}^{C_t},$$

$$W_{\text{out}}^+ = \begin{bmatrix} W_{\text{out}} & W_{\text{out}} \end{bmatrix} \in \mathbb{R}^{C_t \times 2C_v}, \quad b_{\text{out}}^+ = \begin{bmatrix} b_{\text{out}} \\ b_{\text{out}} \end{bmatrix} \in \mathbb{R}^{2C_v}. \tag{5}$$

This initialization ensures that the depth modality is initially a zero-influence pathway, allowing the model to begin fine-tuning from the pretrained RGB state without disrupting performance. During fine-tuning, the model progressively learns to represent depth channels.

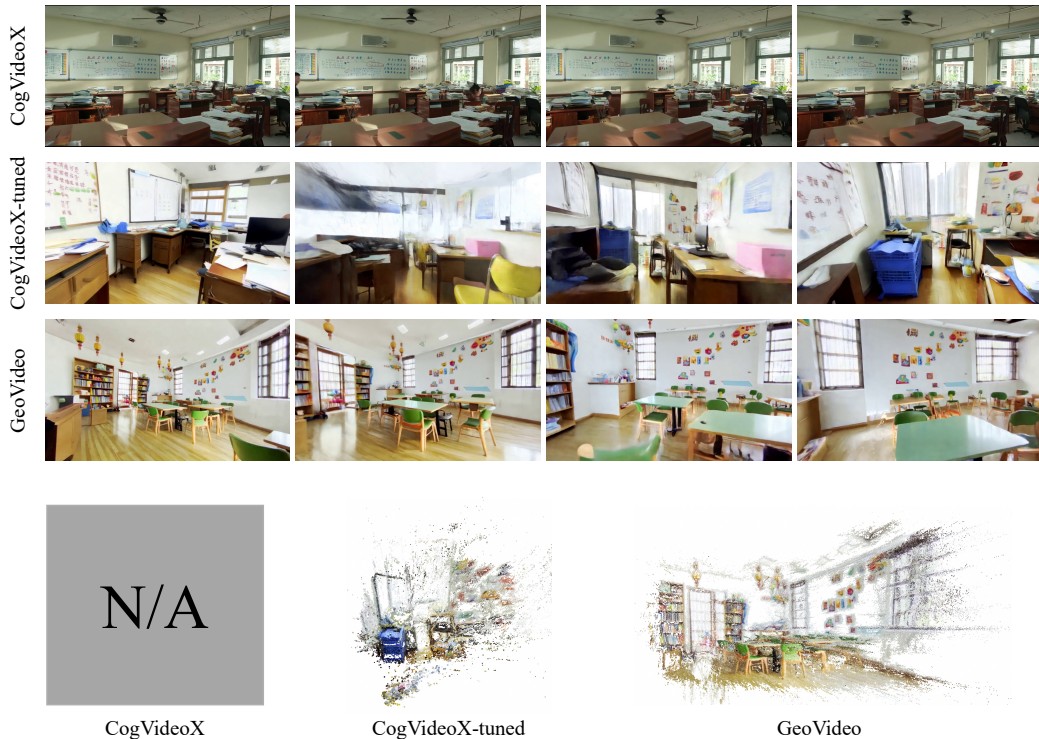

Figure 3: **Comparisons on the down-stream 3D reconstruction task.** The detailed prompt inputs are provided in the supp. materials. The top three rows show samples of video generation results of each method, while the corresponding 3D reconstructions are show in the bottom row.The videos generated by our method offer complete and high-quality 3D reconstructions.

**Stage 1: RGB-D Joint Generation.**    In the first stage, we fine-tune the model to generate RGB and depth frames simultaneously. To ensure stable learning, we apply a gradually increasing weight to the depth loss:

$$\lambda_{\text{depth}}(t) = \min(1.0, 0.1 + \alpha t), \tag{6}$$

where $t$ is the training step and $\alpha$ is set to 0.0001. This gradual increase allows the model to adapt to the new depth supervision without destabilizing RGB generation. The loss for this stage is:

$$\mathcal{L}_{\text{stage-1}} = \mathcal{L}_{\text{diff}}^{\text{RGB}} + \lambda_{\text{depth}}(t) \cdot \mathcal{L}_{\text{diff}}^{\text{D}}. \tag{7}$$

Here, $\mathcal{L}_{\text{diff}}^{\text{RGB}}$ and $\mathcal{L}_{\text{diff}}^{\text{D}}$ are formulated using the v-prediction [48] strategy.

**Stage 2: Geometric Regularization.**    Once the model can generate perceptually and structurally coherent depth maps, we introduce the geometric regularization term $\mathcal{L}_{\text{geo}}$ (described in Section 3). This loss encourages cross-frame depth consistency via reprojection-based supervision.

The final training objective becomes:

$$\mathcal{L}_{\text{total}} = \mathcal{L}_{\text{diff}}^{\text{RGB}} + \lambda_{\text{depth}} \cdot \mathcal{L}_{\text{diff}}^{\text{D}} + \lambda_{\text{geo}} \cdot \mathcal{L}_{\text{geo}}. \tag{8}$$

Here, $\lambda_{\text{geo}}$ is set to 0.5. This staged training process allows the model to gradually learn to incorporate geometry without sacrificing visual fidelity.

## 4    Experimental Results

**Implementation details.** Our experiments are primarily based on CogVideoX-5B [69], a popular and advanced diffusion-based video generation model built on the DiT architecture. We conducted experiments for both text-to-video (T2V) and image-to-video (I2V) generation tasks. The experiments

Table 1: **Multi-view geometric consistency evaluation on the DL3DV dataset.** We use VGGT to predict multi-frame depth and pose, and evaluate consistency using our proposed Multi-View Consistency Score (MVCS) and reprojection error. MVCS measures frame-to-frame depth consistency, ↑: higher is better; while Reprojection Error evaluates how well the globally reconstructed 3D structure aligns with original views, ↓: lower is better.

| Method | MVCS ↑ | Reproj. Error ↓ |
|---|---|---|
| CogVideoX-5B (T2V) | 61.2 | 4.58 |
| CogVideoX-5B (T2V, finetuned on DL3DV) | 66.4 | 3.91 |
| Ours w/o $\mathcal{L}_{\text{geo}}$ (T2V) | 71.3 | 3.36 |
| **Ours (T2V)** | **77.2** | **2.52** |

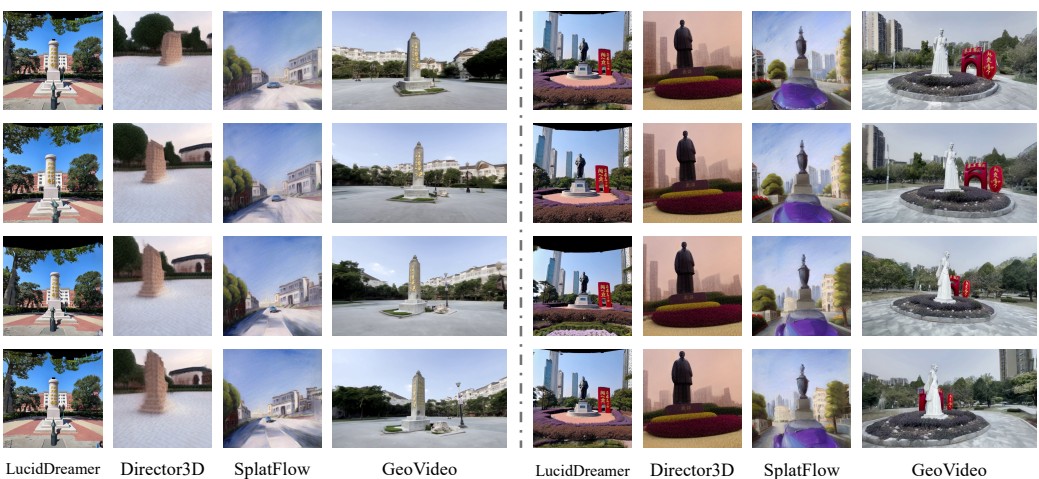

| LucidDreamer | Director3D | SplatFlow | GeoVideo | LucidDreamer | Director3D | SplatFlow | GeoVideo |

Figure 4: **Comparisons with text to 3d scene generation methods.** The detailed prompt inputs are provided in the supp. materials.

are categorized into two types: videos of static scenes and videos of dynamic scenes. For static scenes, we train on the DL3DV-10K [33] dataset. For dynamic videos, we collect a large-scale dataset of approximately 200,000 videos from online sources such as Pexels [40]. We use 544 and 1000 videos from the two datasets for evaluation, respectively. We train the model with a learning rate of 2e-5 on 8 H20 GPUs for 20,000 steps. The batch size is 1 per GPU, with 15K steps for stage 1 and 5K steps for stage 2. The video resolution is set to 768×1360 with 81 frames, following the standard configuration supported by CogVideoX. The depth labels for videos are estimated using Video Depth Anything [7], while for dynamic videos, we estimate camera poses using MegaSaM [31]. For video captions, we use CogVLM [60], as adopted in CogVideo. $\mathcal{D}_p$, $\mathcal{D}_c$, and $\mathcal{D}_d$ are distilled from the corresponding outputs of MegaSaM [31], VGGT [58], and Video Depth Anything [7], respectively. After distillation, their parameters are kept fixed during fine-tuning to provide supervision.

**Scene generation results.** The original video generation model already possesses some ability to generate coherent scenes. However, due to the typically small range of viewpoint changes, it is difficult to extract sufficient multi-view information from the generated videos to reconstruct a meaningful 3D scene (as shown in the first row of Figure 3). To specialize the model for scene-level video generation, we fine-tune the video generation model on the DL3DV dataset [33]. However, since the base model relies purely on 2D modeling, it struggles to maintain geometric consistency under rapid camera viewpoint changes. The second row of Figure 3 shows the results of CogVideoX after being fine-tuned on DL3DV, where such inconsistencies are still apparent. In contrast, after integrating our proposed geometric modeling framework, we are able to maintain consistent multi-view structures even under complex viewpoint variations (Figure 3, third row). To further demonstrate the geometric consistency of the generated results, we perform structure-from-motion (SfM) [49] reconstruction on videos generated by different methods. The original CogVideoX outputs fail for reconstruction due to a lack of sufficient multi-view cues. The fine-tuned model only achieves partial

Table 2: **Quantitative results on the DL3DV dataset** for text-to-3D scene generation. We compare our method against Director3D, LucidDreamer, and SplatFlow across multiple perceptual metrics. ↑: higher is better; ↓: lower is better.

| Method | FID ↓ | CLIPScore ↑ | NIQE ↓ | BRISQUE ↓ |
|---|---|---|---|---|
| LucidDreamer [9] | 79.96 | 31.25 | 11.23 | 44.52 |
| Director3D [30] | 90.20 | 30.04 | 13.79 | 51.67 |
| SplatFlow [11] | 86.77 | 31.42 | 14.15 | 48.85 |
| **Ours** | **72.78** | **33.84** | **8.53** | **36.41** |

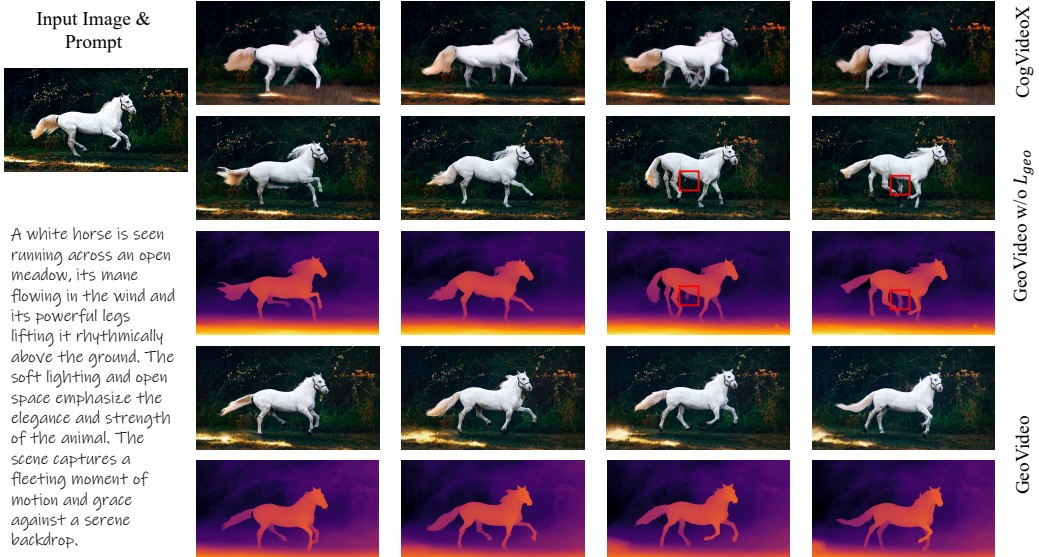

Figure 5: **Comparison with original video generation method & ablation study.** The original video generation model, as well as naive joint modeling of depth, struggle to maintain geometric consistency of objects throughout their motion.

reconstruction with limited consistency among a few consecutive frames. In contrast, our method enables high-quality and structurally complete 3D scene reconstructions.

To further evaluate 3D consistency across frames, we introduce two metrics based on VGGT [58]: the **Multi-View Consistency Score (MVCS)** and the **Reprojection Error**. Given the predicted depth maps and camera poses from VGGT, we project each frame into a shared 3D space to compute cross-view consistency. MVCS measures the alignment of depth maps across views by warping each depth map to neighboring frames and comparing it against the corresponding predicted depth. In contrast, Reprojection Error evaluates the pixel-wise distance between original image coordinates and those reprojected from the reconstructed global point cloud, serving as a direct indicator of geometric alignment accuracy. Table 1 reports MVCS and Reprojection Error for different models on the DL3DV dataset. Our method significantly outperforms CogVideoX-5B and its finetuned variant. Notably, removing the geometric loss $\mathcal{L}_{\text{geo}}$ leads to a clear performance drop, demonstrating the effectiveness of our geometric regularization in preserving 3D structural consistency across views.

Furthermore, we compare our method with three representative text-to-3D scene generation baselines: **Director3D** [30], **LucidDreamer** [9], and **SplatFlow** [11]. The qualitative comparison of generation results can be found in Figure 4. We then follow Director3D and use the following metrics for quantitative evaluation. We use **CLIPScore** [15] to assess the alignment between the generated content and the text prompt. For perceptual evaluation, we adopt the **Fréchet Inception Distance (FID)**[16], a standard metric for assessing visual fidelity and diversity. In addition, we use two *no-reference image quality metrics*—**Natural Image Quality Evaluator (NIQE)**[37] and **Blind/Referenceless Image Spatial Quality Evaluator (BRISQUE)** [36] to directly assess perceptual quality based on image statistics, without relying on ground truth references. Table 2 summarizes the quantitative results in the DL3DV dataset. Our method consistently outperforms all baselines in all metrics. In particular, we achieve the lowest FID and the highest CLIPScore, demonstrating superior semantic consistency

Table 3: **Quantitative comparison with CogVideoX-5B on T2V and I2V tasks.** We evaluate on **CLIPScore**, **FVD** (Fréchet Video Distance), and metrics from **VBench**: **Subject Consistency (SC)**, **Background Consistency (BC)**, **Motion Smoothness (MS)**, **Spatial Relationship (SR)**, and **Video-Image Subject Consistency (VISC)**. ↑: higher is better; ↓: lower is better.

| Method | CLIPScore ↑ | FVD ↓ | SC ↑ | BC ↑ | MS ↑ | SR ↑ | VISC ↑ |
|---|---|---|---|---|---|---|---|
| CogVideoX-5B (T2V) | 32.30 | 145.3 | 93.8 | 95.1 | 93.2 | 79.4 | - |
| Ours w/o $\mathcal{L}_{geo}$ (T2V) | 33.25 | 134.2 | 94.3 | 96.0 | 95.4 | 87.4 | - |
| Ours w/o MP (T2V) | 33.83 | 131.6 | 95.8 | 96.3 | 96.7 | 88.2 | - |
| **Ours (T2V)** | **34.25** | **122.7** | **97.2** | **97.8** | **98.1** | **90.3** | - |
| CogVideoX-5B (I2V) | 33.42 | 139.8 | 94.6 | 96.4 | 95.9 | 80.5 | 95.2 |
| Ours w/o $\mathcal{L}_{geo}$ (I2V) | 34.13 | 128.0 | 95.0 | 97.1 | 96.3 | 86.3 | 96.3 |
| Ours w/o MP (I2V) | 34.77 | 126.5 | 96.9 | 97.6 | 96.9 | 88.8 | 96.8 |
| **Ours (I2V)** | **35.02** | **120.5** | **98.1** | **98.5** | **98.6** | **91.1** | **97.6** |

and perceptual quality. Moreover, our NIQE and BRISQUE scores are also lower, indicating a closer match to natural image distributions compared to prior methods. Compared to these 3D generation methods, our approach achieves higher visual fidelity by leveraging the strong priors from pretrained video generation models.

**Video generation results.** Table 3 presents a quantitative comparison between our method and CogVideoX-5B on both text-to-video and image-to-video generation using the 1000 evaluation videos. We report **CLIPScore** [15] to measure semantic alignment, **FVD** [54] to assess overall visual quality and temporal consistency, and multiple metrics from **VBench** [21], including **Subject Consistency (SC)**, **Background Consistency (BC)**, **Motion Smoothness (MS)**, **Spatial Relationship (SR)**, and **Video-Image Subject Consistency (VISC)**. Our method achieves superior performance across all metrics, clearly outperforming CogVideoX-5B in both settings. Notably, we observe substantial improvements in motion quality (MS), subject coherence (SC), and Spatial Relationship (SR).

**Ablation Studies.** Our method is based on two core components: (1) explicitly modeling depth and (2) applying supervision on the jointly generated depth. We conduct ablation studies targeting these two aspects. As shown in Figure 5, we present an image-to-video (I2V) result demonstrating that simply modeling both depth and appearance already improves the video quality to some extent. This is because the temporal consistency of the depth labels themselves imposes a form of constraint across frames. However, this alone is insufficient to ensure global geometric consistency throughout the video. When our proposed geometric loss $\mathcal{L}_{geo}$ is added, the generated videos exhibit significantly improved frame-to-frame continuity and structural coherence. The corresponding metric improvements are shown in Table 3 and Table 1. In addition, we also study the impact of incorporating the motion probability (MP) map in dynamic videos. Table 3 shows that ignoring the MP map leads to noticeable performance drops, particularly in the T2V setting.

## 5 Conclusions

In this work, we proposed to augment pre-trained video generation models with *geometric regularization* by introducing a per-frame depth prediction and enforcing cross-frame depth consistency. This approach leverages the natural spatio-temporal cues encoded in video to guide the model toward learning stable and physically grounded scene representations. Our method provides implicit geometric supervision through depth alignment, allowing for more accurate modeling of object permanence, spatial relationships, and scene dynamics. Through extensive experiments, we demonstrate that our framework significantly improves the geometric fidelity and temporal stability of generated videos, outperforming prior baselines in both qualitative and quantitative evaluations. We believe that this represents an important step toward bridging video generation and 3D world modeling, opening new possibilities for downstream applications in simulation, robotics, and embodied intelligence.

**Acknowledgements.** This work was supported by Damo Academy through Damo Academy Innovative Research Program. This research has been supported by computing support on the Vista GPU Cluster through the Center for Generative AI (CGAI) and the Texas Advanced Computing Center (TACC) at UT Austin.

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
