# OpenReview forum: "GeoVideo: Introducing Geometric Regularization into Video Generation Model"
_NeurIPS.cc/2025/Conference — NeurIPS 2025 poster_

### Official Review · Reviewer_jMr7 · 2025-06-28

**Clarity:** 3
**Significance:** 2
**Originality:** 3
**Rating:** 4
**Confidence:** 4

**Summary:**

this paper proposes a method to predict depth video, camera parameters, and motion probability along with the rgb videos. by using the additional information, an additional loss term that measures the 3d consistency is defined and applied, thus improving the 3d consistency of the generated video.

**Questions:**

1. [critical] please provide more details on the 1000 test videos in Table 3, which i think is very important but is missing.
2. [critical] please include more qualitative results to show the effectiveness of proposed method, with various inputs. the cases shown in the paper (as in Figure 3,4,5) are over simple for a practical video generation applications.
3. [less critical] why does not perform a complete vbench evaluation benchmark? i am curious about the performance of other metrics.
4. [less critical] i notice that in Table 1 and 3, the performance of "ours w/o L_geo" is already better than the baseline. is there any case where "ours w/o L_geo" performs better than "ours"? is there any case where the baseline performs better than "ours w/o L_geo"?

**Ethical Concerns:**

["NO or VERY MINOR ethics concerns only"]

**Final Justification:**

the latest rebuttal resolved most of my concerns, and i raised my score accordingly.

**Limitations:**

[critical] the Limitation section is quite brief. please include the negative influence of apply such a artificial loss term.

**Quality:**

2

**Strengths And Weaknesses:**

strengths:
1. the idea is quite simple and straightforward, and the paper is easy to read.

weaknesses:
1. the experiments are less convincing. for example, the authors collected a dataset for training and evaluation for dynamic video generation. as the data has a critical influence on the experimental results, i am curious about the fairness of the experimental results as shown in Table 3.
2. forcing such an loss may introduce some negative influences on the generation performance, but there is no such analyses.

as i have major concerns on the experiments, i would suggest borderline reject for now.

---

> ### Author Rebuttal · Authors · 2025-07-29
>
> **Thank you very much for your time and valuable insights. We truly appreciate your recognition of certain aspects of our work—it is highly encouraging. Please find our responses to your comments below.**
>
>    1.**Details about the test videos.**
>    Due to the lack of high-resolution dynamic scenes in most public video datasets, we collected video data from websites such as Pexels for our experiments. It is difficult for us to list the exact source of each individual video, but we can roughly provide the distribution of the test videos as follows.
>
> | Main Category        | Subcategory                            | % of Total | Notes                                      |
> | -------------------- | -------------------------------------- | ---------- | ------------------------------------------ |
> | Entertainment        | Music videos / performances            | 7.3%       | Official MVs, covers, busking, instruments |
> |                      | Comedy skits /edits                    | 6.6%       | Internet humor, reactions         |
> |                      | Gaming streams / montages              | 3.7%       | FPS, Minecraft, mobile games               |
> |                      | Movie clips / fan edits / reviews      | 6.4%       | Clips, trailers, commentary                |
> | People & Lifestyle   | Daily vlogs / personal storytelling    | 6.4%       | Travel, routines, life updates             |
> |                      | Social challenges / micro-dramas       | 3.8%       | Pranks, street interviews, short skits     |
> |                      | Beauty / fashion / routines            | 4.1%       | Makeup, hauls, get-ready-with-me           |
> |                      | Family / pets / home life              | 5.4%       | Baby clips, couples, dogs and cats         |
> | Education & Tech     | How-to tutorials (tech/art/DIY)        | 5.8%       | DIY, painting, home repairs                |
> |                      | Science / explainer videos             | 2.9%       | Space, biology, psychology                 |
> |                      | Product reviews / unboxings            | 2.6%       | Phones, gear, tech demos                   |
> | Nature & Travel      | Landscape / drone                      | 5.4%       | Mountains, oceans, cityscapes              |
> |                      | Underwater / macro / slow motion       | 2.9%       | Fish, ice, flowers, insects                |
> |                      | Wildlife / animal behavior             | 4.9%       | Birds, tigers, elephants, insects          |
> |                      | Road trips / hidden gems               | 2.4%       | Nature trails, rural areas                 |
> | Urban & Architecture | City walking / night photography       | 2.7%       | POV walks, rain, lights                    |
> |                      | Buildings / design / interiors         | 1.8%       | Real estate, modern design                 |
> |                      | Street food / markets / urban culture  | 2.2%       | Food stalls, crowds, culture               |
> | Fitness & Movement   | Yoga / workouts / wellness             | 2.8%       | Gym sessions, meditation                   |
> |                      | Dance / choreography / freestyle       | 2.3%       | K-pop, street, experimental                |
> |                      | Extreme sports / climbing / skating    | 2.1%       | solo stunts, aerials                |
> | Food & Cooking       | Cooking demos / recipes                | 2.9%       | Fried rice, baking, plating                |
> |                      | Mukbang / ASMR eating                  | 1.7%       | Eating sounds, visual food focus           |
> | Business / Work      | Ad campaigns / product shots           | 2.9%       | Branded short ads, slow-motion products    |
> |                      | Offices / team meetings / interviews   | 1.3%       | Corporate scenes, recruitment              |
> |                      | Exhibition videos                      | 1.4%       | Real estate/auto show videos               |
> | Religion / Rituals   | Weddings / festivals / spiritual clips | 2.2%       | Church, temples, cultural celebrations     |
> | Other / Niche        |                                        | 3.1%       |                                            |
>
>
>
>    2. **More qualitative results.**
>    While you may have already noticed this, we would still like to kindly remind you that, in addition to the figures shown in the main paper, we have also provided additional video results via an **anonymous link in the supplementary material.** Due to NeurIPS's rebuttal policy, we are not allowed to provide further visualizations at this stage, but the results in the supplementary material do include some complex scene cases.
>
>    3. **Complete VBench metrics.**
>    Since many metrics in VBench are not directly related to geometric properties, we primarily showcase those that are geometry-related in the main paper. Here, we provide the complete set of metrics for reference. If necessary, we will include these metrics in the revised version of the paper.
>
>    | Models        | Subject Consistency | Background Consistency | Temporal Flickering | Motion Smoothness | Dynamic Degree | Aesthetic Quality | Imaging Quality | Object Class |
>    | ------------- | ------------------- | ---------------------- | ------------------- | ----------------- | -------------- | ----------------- | --------------- | ------------ |
>    | CogVideoX     | 93.78%              | 95.08%                 | 98.73%              | 93.17%            | 66.13%         | 61.97%            | 64.74%          | 85.12%       |
>    | Ours          | 97.22%              | 97.79%                 | 99.46%              | 98.11%            | 87.74%         | 63.89%            | 68.22%          | 94.34%       |
>
>    | Models        | Multiple Objects | Human Action | Color   | Spatial Relationship | Scene  | Appearance Style | Temporal Style | Overall Consistency |
>    | ------------- | ---------------- | ------------ | ------- | -------------------- | ------ | ---------------- | -------------- | ------------------- |
>    | CogVideoX     | 69.88%           | 96.57%       | 87.30%  | 79.43%               | 51.17% | 22.38%           | 25.38%         | 27.12%              |
>    | Ours          | 85.93%           | 99.20%       | 90.83%  | 90.33%               | 61.42% | 22.53%           | 25.82%         | 28.21%  |
>
> 4. **Cases where geometric regularization performs worse.**
>    Such cases may exist, for example, as mentioned in our limitation section, for phenomena like fire and smoke. These objects can *somehow* be estimated with depth, and are therefore modeled with corresponding depth in our generation process. However, the addition of the geometric loss term $L_{geo}$ might reduce the dynamic characteristics of such videos compared to *ours w/o* $L_{\text{geo}}$. These non-rigid, volumetric phenomena may require more sophisticated mechanisms, which are currently beyond the scope of this study. Due to current constraints, we are unable to provide visualizations and can only describe the results verbally. We appreciate your understanding. Nevertheless, based on the average quantitative metrics across most scenarios, our geometric regularization still **shows improvement for the majority of video generation scenarios.**
>
>    In most cases, *ours w/o* $L_{\text{geo}}$ already outperforms baseline methods. This is because it introduces additional channels with minimal interference to the original pretrained model. Even when depth is not semantically essential to the video content, modeling depth as auxiliary information can still enhance temporal consistency, as depth labels exhibit strong temporal coherence.
>
>
> 5. **More comprehensive limitations.** While our method incorporates geometric reasoning to improve the consistency of video generation, it still exhibits some limitations. First, the concept of "video" spans a wide range of visual content, and our approach struggles in scenarios where geometry is ill-defined or not meaningful—such as smoke, fire, reflections, or 2D animations—which often leads to suboptimal outputs. Second, although the geometric loss term helps enforce spatial coherence, it may also impose rigid constraints that suppress plausible appearance changes or motion patterns inconsistent with the assumed geometry. This can reduce the diversity or realism of generated results in cases that fall outside standard 3D assumptions. Lastly, the model may not generalize well to out-of-distribution inputs, particularly those involving visual phenomena that cannot be effectively captured by depth-based representations. Addressing these challenges may require more flexible or content-aware mechanisms that can selectively apply geometric priors based on scene characteristics.

---

> > ### Comment · Reviewer_jMr7 · 2025-08-04
> >
> > thanks for the rebuttal. i still have concerns about the test video dataset.
> >
> > the authors mentioned "Due to the lack of high-resolution dynamic scenes in most public video datasets, we collected video data from websites such as Pexels for our experiments", which i do not think is a sufficient reason to introduce a new and closed-source dataset for evaluation. as the author mentioned, there are limitations on the proposed method when the domain of data violates the key assumptions of 3d consistency, that means the dataset for evaluation is crucial for the performance of the proposed method.

---

> > > ### Author Response · Authors · 2025-08-04
> > >
> > > Thank you very much for your active engagement in the discussion! We would like to point out that, in addition to dynamic videos, our method also performs well on open-source static scene dataset such as DL3DV. Beyond VBench, and in response to suggestions from other reviewers, we have also included evaluations using **WorldScore** [1], a newly proposed open-source benchmark (please refer to our other rebuttal). We believe WorldScore is better aligned with the evaluation needs of our method. After testing on their provided data, our results are summarized below. As shown, our method achieves noticeable improvements over comparable baselines, particularly in terms of **3D consistency**. We will include this benchmark in the revised version of our paper.
> > >
> > > While real-world video generation is inherently complex and it is unrealistic for any single method to handle all possible cases, we argue that the **assumption of 3D consistency** is broadly applicable and beneficial across the majority of generation scenarios.
> > >
> > > Furthermore, to contribute to the community, we will release our code and pretrained weights.
> > >
> > > Thank you again for your thoughtful response!
> > >
> > >
> > > | Models            | WorldScore-Static | WorldScore-Dynamic | Camera Ctrl | Object Ctrl | Content Align | 3D Consist | Photo Consist | Style Consist | Subjective Qual | Motion Acc | Motion Mag | Motion Smooth |
> > > |-------------------|-------------------|---------------------|--------------|--------------|----------------|--------------|----------------|----------------|------------------|-------------|-------------|----------------|
> > > | ViewCrafter       | 45.01             | 11.01               | 88.26        | 48.52        | 58.45          | 78.10        | 69.13          | 66.52          | 59.93            | 2.38        | 1.08         | 17.00          |
> > > | AC3D-5B           | 51.14             | 45.12               | 91.01        | 48.14        | 65.08          | 65.48        | 60.61          | 38.61          | 41.17            | 22.34       | 43.74        | 32.41          |
> > > | DimensionX        | 59.58             | 47.63               | 93.27        | 39.52        | 36.33          | 82.04        | 82.44          | 81.08          | 61.33            | 63.63       | 25.59        | 54.76          |
> > > | CogVideoX-T2V     | 54.18             | 48.79               | 40.22        | 51.05        | 68.12          | 68.81        | 64.20          | 42.19          | 44.67            | 25.00       | 47.31        | 36.28          |
> > > | Ours-T2V          | 57.38             | 52.03               | 42.84        | 55.49        | 70.61          | 90.88        | 73.64          | 49.71          | 48.88            | 28.66       | 50.63        | 39.33          |
> > > | CogVideoX-I2V     | 62.15             | 59.12               | 38.27        | 40.07        | 36.73          | 86.21        | 88.12          | 83.22          | 62.44            | 69.56       | 26.42        | 60.15          |
> > > | Ours-I2V          | 64.57             | 62.03               | 41.02        | 42.41        | 38.45          | 92.11        | 90.15          | 86.10          | 64.38            | 72.02       | 29.10        | 62.85          |
> > >
> > >
> > > [1] Duan H, Yu H X, Chen S, et al. Worldscore: A unified evaluation benchmark for world generation[J]. arXiv preprint arXiv:2504.00983, 2025.

---

### Official Review · Reviewer_6j7J · 2025-07-01

**Clarity:** 3
**Significance:** 2
**Originality:** 2
**Rating:** 4
**Confidence:** 4

**Summary:**

This manuscript introduces geometric regularization into latent video diffusion models to improve spatiotemporal consistency.
Building on a standard DiT backbone, the authors add parallel depth and camera-pose prediction branches so that each generated RGB frame is accompanied by a depth map and estimated camera pose.
During training, they first jointly learn RGB and depth generation, then in a second stage enforce a geometric consistency loss: depth predictions from different frames are back-projected into a shared 3D point cloud and re-projected to each view, penalizing depth misalignment across frames.
This two-stage fine-tuning yields videos whose frames not only look realistic but also adhere to a coherent underlying 3D structure. Extensive experiments on both static (DL3DV) and dynamic video datasets show marked improvements in multi-view depth consistency, reprojection error, downstream SfM reconstruction success, and perceptual/video-quality metrics.

**Questions:**

please refer to weaknesses.

**Ethical Concerns:**

["NO or VERY MINOR ethics concerns only"]

**Final Justification:**

The authors have solved most of my concern, I stick to the oginiral rating.

**Limitations:**

yes

**Quality:**

3

**Strengths And Weaknesses:**

Strengths:
1.By generating depth and poses alongside RGB and enforcing cross-frame depth alignment, GeoVideo addresses temporal shape flicker and structural drift common in 2D-only methods.

2. The manuscript reports improvements on multi-view consistency scores, re-projection error, 3D reconstruction success, and standard perceptual/video metrics, demonstrating both geometric and visual gains.

3. Separating joint RGB-depth learning from geometric regularization stabilizes training and prevents early geometric constraints from harming visual quality.


Weaknesses:
1. The idea of simultaneously generating RGB frames and geometric cues like depth is not entirely new—earlier work such as [1] has already explored joint synthesis of appearance and geometry (e.g., point maps).

2. The evaluation omits several state-of-the-art text-to-3D and video generation methods (e.g., AC3D, Viewcrafter), so it is unclear how GeoVideo compares against the full spectrum of recent 3D-aware approaches.

3. Relying solely on VBench metrics assesses perceptual and temporal coherence but does not directly evaluate 3D world fidelity; incorporating WorldScore [2]  would provide a more complete picture of GeoVideo’s reconstruction capabilities.


[1] Zhang Q, Zhai S, Martin M A B, et al. World-consistent video diffusion with explicit 3d modeling[C]//Proceedings of the Computer Vision and Pattern Recognition Conference. 2025: 21685-21695.
[2] Duan H, Yu H X, Chen S, et al. Worldscore: A unified evaluation benchmark for world generation[J]. arXiv preprint arXiv:2504.00983, 2025.

---

> ### Author Rebuttal · Authors · 2025-07-29
>
> **Thank you very much for your time and valuable insights. We truly appreciate your recognition of certain aspects of our work—it is highly encouraging. Please find our responses to your comments below.**
> 1. **Difference from WVD.**
>    WVD simply models additional XYZ channels jointly, without introducing any specific regularization on those channels. Moreover, their approach is limited to static scenes, cannot generate dynamic videos, and does not support prompt-based video generation. And WVD does not provide open-source code, making a comprehensive comparison difficult. In our paper, we have cited this work in the *Related Work* section and highlighted the differences. The main contribution of our method lies in introducing **geometric regularization** specifically on the additional modeling channels to enhance consistency in video generation.
>
> 2. **Comparison with AC3D and ViewCrafter.**
>    Both AC3D and ViewCrafter focus primarily on improving the **spatial controllability** of video generation. In contrast, our goal is to enable the video model itself to generate **geometry-grounded** videos, which is fundamentally different. Nevertheless, we can still compare the video quality of our method against these approaches, as shown below. (See the next question — we use WorldScore uniformly for comparison here.) Since these methods introduce explicit camera pose as conditional input, their models exhibit better camera controllability.
>
> 3. **WorldScore Metrics.**
>    WorldScore is a recently proposed benchmark. In fact, some of the metrics we compute—such as MVCS—share the same underlying principles as certain metrics (3D Consist etc.) from WorldScore. Here, we provide a supplementary evaluation of our method on the WorldScore benchmark. If necessary, we will include these metrics in the revised version of the paper.
>
>
>    | Models        | WorldScore-Static | WorldScore-Dynamic | Camera Ctrl | Object Ctrl | Content Align | 3D Consist | Photo Consist | Style Consist | Subjective Qual | Motion Acc | Motion Mag | Motion Smooth |
>    | ------------- | ----------------- | ------------------ | ----------- | ----------- | ------------- | ---------- | ------------- | ------------- | --------------- | ---------- | ---------- | ------------- |
>    | ViewCrafter   | 45.01             | 11.01              | 88.26       | 48.52       | 58.45         | 78.10      | 69.13         | 66.52         | 59.93           | 2.38       | 1.08       | 17.00         |
>    | AC3D-5B       | 51.14             | 45.12              | 91.01       | 48.14       | 65.08         | 65.48      | 60.61         | 38.61         | 41.17           | 22.34      | 43.74      | 32.41         |
>    | DimensionX    | 59.58             | 47.63              | 93.27       | 39.52       | 36.33         | 82.04      | 82.44         | 81.08         | 61.33           | 63.63      | 25.59      | 54.76         |
>    | CogVideoX-T2V | 54.18             | 48.79              | 40.22       | 51.05       | 68.12         | 68.81      | 64.20         | 42.19         | 44.67           | 25.00      | 47.31      | 36.28         |
>    | Ours-T2V      | 57.38             | 52.03              | 42.84       | 55.49       | 70.61         | 90.88      | 73.64         | 49.71         | 48.88           | 28.66      | 50.63      | 39.33         |
>    | CogVideoX-I2V | 62.15             | 59.12              | 38.27       | 40.07       | 36.73         | 86.21      | 88.12         | 83.22         | 62.44           | 69.56      | 26.42      | 60.15         |
>    | Ours-I2V      | 64.57             | 62.03              | 41.02       | 42.41       | 38.45         | 92.11      | 90.15         | 86.10         | 64.38           | 72.02      | 29.10      | 62.85         |

---

### Official Review · Reviewer_8Q1X · 2025-07-01

**Clarity:** 3
**Significance:** 3
**Originality:** 3
**Rating:** 4
**Confidence:** 3

**Summary:**

This paper proposes GeoVideo, a geometry-regularized latent video generation framework that integrates explicit geometric consistency constraints into diffusion-based video synthesis. The key idea is to predict depth and camera pose jointly with RGB frames and enforce multi-frame consistency via a novel cross-frame reprojection loss computed from 3D point clouds. The proposed method is lightweight and modular, making it easy to plug into existing latent diffusion architectures such as CogVideoX and DiT.

**Questions:**

1. How robust is the geometric supervision when the predicted camera poses are inaccurate? Have you evaluated sensitivity to pose errors?

2. Can the model handle complex dynamic scenes (e.g., fast motion, occlusions)? How does the motion probability map behave in such cases?

3. Have you considered using external pose/depth priors (e.g., from COLMAP or pretrained estimators) to improve supervision or initialization?

4. How does the proposed method compare quantitatively against WVD or UniGeo, which also integrate geometry into diffusion-based pipelines?

**Ethical Concerns:**

["NO or VERY MINOR ethics concerns only"]

**Final Justification:**

The rebuttal has addressed my concerns on the robustness of geometry supervision, the performance on complex dynamic scenes, questions on external pose/depth priors, and the comparisons with WVD and UniGeo. I am still positive about this paper after reading the other reviews and the rebuttal. Overall, I think this work offers an effective solution to forster structural consistency in video diffusion models.

**Limitations:**

- The geometric regularization is limited to self-predicted geometry, which may not be fully reliable under complex motion, low lighting, or ambiguous structures.

- The method assumes rigid camera motion and does not account for object-level 3D dynamics, which limits applicability to highly dynamic scenes.

- The geometric consistency loss is currently applied only as an auxiliary signal and not deeply coupled into the generation process (e.g., via 3D-aware attention mechanisms or geometry-conditioned denoising).

**Paper Formatting Concerns:**

No issues.

**Quality:**

3

**Strengths And Weaknesses:**

--- Strengths

- Well-motivated integration of geometry: Instead of pursuing full 3D scene modeling (e.g., NeRF-like pipelines), the paper proposes a lightweight, practical mechanism to improve structural consistency in video generation using explicit geometric priors.

- Plug-and-play design: The method requires minimal modification to existing latent diffusion models and can be seamlessly integrated into popular T2V frameworks.

- The cross-frame reprojection loss with motion-aware masking is effective in promoting geometric coherence without requiring ground-truth depth or camera pose.

- The paper reports better results on both traditional perceptual metrics (FID, CLIPScore, NIQE) and structural metrics (MVCS, Reprojection Error). The video visualization in the supplemental materials also demonstrate the effectiveness of the proposed method.

--- Weaknesses

- The method relies on predicted depth and camera pose from the same model, which may introduce circular supervision bias. Errors in pose estimation can propagate into the reprojection loss.

- Limited comparison to other recent geometry-aware diffusion models, such as WVD or UniGeo, especially in ablations or direct quantitative analysis.

- No analysis on failure cases or limits of the geometric consistency under highly dynamic or low-texture regions.

--- Reasons of Rating

- Overall, GeoVideo offers a practical, effective, and relatively novel solution to enhancing structural consistency in video diffusion models. While it does not radically redefine the field, its design simplicity, compatibility, and demonstrated gains make it a meaningful contribution. Strengthening comparative analysis and discussing failure cases would significantly increase its impact. Hence, I'm positive for this work.

---

> ### Author Rebuttal · Authors · 2025-07-30
>
> **Thank you very much for your time and valuable insights. We truly appreciate your recognition of certain aspects of our work—it is highly encouraging. Please find our responses to your comments below.**
> 1. **Robustness of geometric supervision.**
>    Yes, our approach can be influenced by the accuracy of the various prediction heads. Although components like our camera head already yield reasonably accurate predictions **(see our response to Question 3)**, the outputs from different heads are inherently challenging to predict with perfect precision. To improve the robustness of our model, we apply geometric regularization **only at locations where the difference between the reprojected depth and the generated depth is sufficiently small**.
>    We can include an ablation showing that removing this tolerance threshold leads to a degradation in generation quality, mainly due to noise from pose errors and other sources. Introducing a simple tolerance threshold proves to be an effective strategy that reveals the strength of our method.
>
>    | Method                   | CLIPScore ↑ | FVD ↓     | SC ↑     | BC ↑     | MS ↑     | SR ↑     | VISC ↑   |
>    | ------------------------ | ----------- | --------- | -------- | -------- | -------- | -------- | -------- |
>    | CogVideoX-5B (T2V)       | 32.30       | 145.3     | 93.8     | 95.1     | 93.2     | 79.4     | -        |
>    | Ours w/o tolerance (T2V) | 32.41       | 141.6     | 94.6     | 95.0     | 95.4     | 86.6     | -        |
>    | **Ours (T2V)**           | **34.25**   | **122.7** | **97.2** | **97.8** | **98.1** | **90.3** | -        |
>    |                          |             |           |          |          |          |          |          |
>    | CogVideoX-5B (I2V)       | 33.42       | 139.8     | 94.6     | 96.4     | 95.9     | 80.5     | 95.2     |
>    | Ours w/o tolerance (I2V) | 33.18       | 136.5     | 95.2     | 96.0     | 95.6     | 87.6     | 95.4     |
>    | **Ours (I2V)**           | **35.02**   | **120.5** | **98.1** | **98.5** | **98.6** | **91.1** | **97.6** |
>
>
>
> 2. **Effects of motion probability maps on complex dynamic scenes.**
>    Due to NeurIPS rebuttal policy, we are not allowed to provide visualizations of the motion probability maps during the rebuttal period. As our motion probability head is distilled from MegaSAM, the predictions closely resemble those of MegaSAM. Their motion probability maps have been shown to improve pose accuracy during bundle adjustment in complex dynamic scenes. In our setup, incorporating such motion predictions also helps improve geometric consistency in generated results for complex dynamic scenarios. We believe the dynamic scenes used in our experiments already present a sufficient level of complexity to demonstrate the effectiveness of our method. Quantitative comparison results for dynamic scenes are presented in the main paper, and additional videos illustrating complex dynamic scenes are included in the supplementary material via an anonymous link.
>
> 3. **Using external pose/depth priors.**
>    Yes, in fact, we have already used pretrained estimators! Our camera pose decoder is distilled from pretrained estimators (VGGT) and learns to predict poses from generated latents. Here, we add a comparison between our predicted poses from video latents and those estimated by VGGT on two datasets—CO3Dv2 and RealEstate10K—using the standard AUC@30 metric, which combines Relative Rotation Accuracy (RRA) and Relative Translation Accuracy (RTA).  Since our camera head is distilled from VGGT, it achieves comparable accuracy to the original VGGT estimator. Our depth head and motion probability head are also distilled from pretrained estimators—VideoDepthAnything and MegaSAM, respectively—and their predictions achieve accuracy comparable to the original estimators. As for COLMAP, it is too time-consuming to be run jointly with video generation, making it impractical for our pipeline. Moreover, its accuracy is lower than that of the pretrained estimator VGGT we use for distillation.
>
>    | Methods             | Re10K (unseen) AUC@30 ↑ | CO3Dv2 AUC@30 ↑ |
>    | ------------------- | ----------------------- | --------------- |
>    | Colmap+SPSG         | 45.2                    | 25.3            |
>    | DUST3R              | 67.7                    | 76.7            |
>    | VGGT (Feed-Forward) | 85.3                    | 88.2            |
>    | Ours (Feed-Forward) | 84.9                    | 88.9            |
>
>
>
> 4. **Comparison with WVD and UniGeo.**
>    WVD does not provide open-source code, making a comprehensive comparison difficult. Their method only performs joint modeling of additional XYZ channels, without applying any explicit regularization on them. Moreover, their approach is limited to static scenes, cannot generate dynamic videos, and does not support prompt-based video generation. We have cited their work in the *Related Work* section and discussed these differences. Here, we report a comparison on depth estimation metrics as presented in their paper for the NYU, BONN, and ScanNet datasets.
>
>     UniGeo, on the other hand, is a fundamentally different type of method. It focuses on estimating geometry from existing video inputs, while our work is centered on video generation. Although UniGeo has a GitHub repository, they do not provide their model code or pretrained weights, making direct comparison difficult. We instead refer to the depth estimation metrics reported in their paper on the 7Scenes dataset.
>     It is also worth noting that the UniGeo paper was released on arXiv on May 30, 2025—after the NeurIPS submission deadline.
>
>     To enable a fair comparison with the above methods in terms of depth estimation, we made a specific modification to our framework: we concatenate a real video as conditional input with the noise, and let our model reconstruct both the video and the corresponding depth. In this way, our model can function to some extent as a depth estimator. However, it is important to emphasize that our method was **not originally designed for geometry estimation**, but rather for **geometry-aware video generation**. Nevertheless, as shown in the comparison tables below, our method also demonstrates a advantage in depth estimation. That said, with the help of explicit supervision, our model can also learn to predict depth from video input effectively.
>
>
>
>    | Methods | NYU-v2 |         | BONN  |         | ScanNet++ |         |
>    | ------- | ------ | ------- | ----- | ------- | --------- | ------- |
>    |         | Rel ↓  | δ₁.₂₅ ↑ | Rel ↓ | δ₁.₂₅ ↑ | Rel ↓     | δ₁.₀₃ ↑ |
>    | WVD     | 9.7    | 90.8    | 7.0   | 96.4    | 5.0       | 57.2    |
>    | Ours    | 6.1    | 97.2    | 6.8   | 96.7    | 3.2       | 80.6    |
>
>    | Methods | 7Scenes    |      |
>    | ------- | -------- | ---- |
>    |         | AbsRel ↓ | δ1 ↑ |
>    | UniGeo  | 20.69    | 62.2 |
>    | Ours    | 18.42    | 71.1 |

---

> > ### Comment · Reviewer_8Q1X · 2025-08-06
> >
> > Thanks for the rebuttal! The rebuttal has addressed my concerns.
> > I would keep my original rating.

---

### Official Review · Reviewer_ee6U · 2025-07-02

**Clarity:** 2
**Significance:** 3
**Originality:** 2
**Rating:** 4
**Confidence:** 4

**Summary:**

This paper aims to predict a high-quality 3D reconstruction by introducing geometric regularization into video diffusion models. Specifically, the authors augment VDM with depth prediction, which serves as an implicit form of geometric supervision. In addition, the paper proposes an explicit geometric regularization term that enhances the geometric consistency of the underlying shapes. The proposed approach is validated through experiments on both text-to-video (T2V) and image-to-video (I2V) tasks.

**Questions:**

The main issues lie in unclear training details (denoising steps) and insufficient experimental comparisons (3D consistency); for details, see the weaknesses.

**Ethical Concerns:**

["NO or VERY MINOR ethics concerns only"]

**Final Justification:**

The reply has addressed my concern. I would like to stay positive rating.

**Limitations:**

yes

**Quality:**

3

**Strengths And Weaknesses:**

**Strengths**
1. The proposed geometric regularization provides explicit supervision for video diffusion models, significantly improving both 3D consistency and temporal coherence in generated videos.

2. The paper also introduces an effective model initialization strategy and a multi-stage training framework, which contribute to stable optimization and improved performance.

**Weaknesses**

[MAJOR]:

1. Diffusion models typically require multiple denoising steps to produce clean outputs. During training with geometric regularization, does the model compute the regularization from one-step denoising output or many-step denoising output? What is the rationale behind this design choice?

2. While the method primarily emphasizes 3D consistency in generated videos, a more comprehensive comparison with state-of-the-art approaches like ViewCrafter[1] and DimensionX[2]—specifically in terms of 3D consistency—would further validate its advantages.

[Minor]

3. Is there any quantitative evaluation of the accuracy or performance consistency across different prediction heads? Such an analysis would better demonstrate whether the model effectively leverages geometric regularization.

4. There are several writing errors. For instance, in line 209, “estimate camera poses using MegaSaM” appears to be incorrect.

[1] Yu W, Xing J, Yuan L, et al. Viewcrafter: Taming video diffusion models for high-fidelity novel view synthesis[J]. arXiv preprint arXiv:2409.02048, 2024.
[2] Sun W, Chen S, Liu F, et al. Dimensionx: Create any 3d and 4d scenes from a single image with controllable video diffusion[J]. arXiv preprint arXiv:2411.04928, 2024.

---

> ### Author Rebuttal · Authors · 2025-07-29
>
> **Thank you very much for your time and valuable insights. We truly appreciate your recognition of certain aspects of our work—it is highly encouraging. Please find our responses to your comments below.**
> 1. **How geometric regularization is incorporated.**
>    At any diffusion step, we can compute the corresponding clean latent representation from the predicted noise using the predefined noise schedule. We apply our geometric regularization directly on computed clean depth latent. Because applying regularization to the clean depth latent is directly effective, and only the clean depth latent can be reliably decoded by our depth decoder into a meaningful depth map. We will provide a more detailed explanation of this mechanism in the revised version of the paper.
> 2. **Comparison with DimensionX and ViewCrafter.**
>    Both DimensionX and ViewCrafter focus primarily on improving the **spatial controllability** of video generation. In contrast, our goal is to enable the video model itself to generate **geometry-grounded** videos, which is fundamentally different. Nevertheless, we can still compare the video quality of our method against these approaches, as shown below. (We use WorldScore [1] uniformly for comparison here.) Since these methods introduce explicit camera pose as conditional input, their models exhibit better camera controllability. If necessary, we will include these comparisions in the revised version of the paper.
>
>     | Models        | WorldScore-Static | WorldScore-Dynamic | Camera Ctrl | Object Ctrl | Content Align | 3D Consist | Photo Consist | Style Consist | Subjective Qual | Motion Acc | Motion Mag | Motion Smooth |
>     | ------------- | ----------------- | ------------------ | ----------- | ----------- | ------------- | ---------- | ------------- | ------------- | --------------- | ---------- | ---------- | ------------- |
>     | ViewCrafter   | 45.01             | 11.01              | 88.26       | 48.52       | 58.45         | 78.10      | 69.13         | 66.52         | 59.93           | 2.38       | 1.08       | 17.00         |
>     | AC3D-5B       | 51.14             | 45.12              | 91.01       | 48.14       | 65.08         | 65.48      | 60.61         | 38.61         | 41.17           | 22.34      | 43.74      | 32.41         |
>     | DimensionX    | 59.58             | 47.63              | 93.27       | 39.52       | 36.33         | 82.04      | 82.44         | 81.08         | 61.33           | 63.63      | 25.59      | 54.76         |
>     | CogVideoX-T2V | 54.18             | 48.79              | 40.22       | 51.05       | 68.12         | 68.81      | 64.20         | 42.19         | 44.67           | 25.00      | 47.31      | 36.28         |
>     | Ours-T2V      | 57.38             | 52.03              | 42.84       | 55.49       | 70.61         | 90.88      | 73.64         | 49.71         | 48.88           | 28.66      | 50.63      | 39.33         |
>     | CogVideoX-I2V | 62.15             | 59.12              | 38.27       | 40.07       | 36.73         | 86.21      | 88.12         | 83.22         | 62.44           | 69.56      | 26.42      | 60.15         |
>     | Ours-I2V      | 64.57             | 62.03              | 41.02       | 42.41       | 38.45         | 92.11      | 90.15         | 86.10         | 64.38           | 72.02      | 29.10      | 62.85         |
>
>
> 3. **Quantitative evaluation of different prediction heads.**
>
>     3.1 **Camera pose head.**
>     We report the results of our camera head, which predicts camera poses from video latents, on two datasets—CO3Dv2 and RealEstate10K—using the standard AUC@30 metric, which combines Relative Rotation Accuracy (RRA) and Relative Translation Accuracy (RTA). We also compare our results with methods such as VGGT. Since our camera head is distilled from VGGT, it achieves comparable accuracy to the original VGGT estimator.
>
>     | Methods             | Re10K (unseen) AUC@30 ↑ | CO3Dv2 AUC@30 ↑ |
>     | ------------------- | ----------------------- | --------------- |
>     | Colmap+SPSG         | 45.2                    | 25.3            |
>     | DUST3R              | 67.7                    | 76.7            |
>     | VGGT (Feed-Forward) | 85.3                    | 88.2            |
>     | Ours (Feed-Forward) | 84.9                    | 88.9            |
>
>
>     3.2 **Motion probability map head.**
>     Since ground truth motion probability maps are not available, it is difficult to directly assess the correctness of the predicted maps. As our motion probability map design is inspired by MegaSAM and the corresponding head is also distilled from it, we report the MSE difference between the outputs of our motion prediction head and those of MegaSAM’s corresponding head on the DyCheck and Sintel benchmarks, to demonstrate that our predictions are qualitatively similar.
>
>     | Datasets    | MSE       |
>     | ---------- | --------- |
>     | DyCheck    | 0.042     |
>     | MPI Sintel | 0.056     |
>
>
>     3.3 **Depth head.**
>     Our depth head learns a mapping from depth latent to actual depthmaps. As the generated videos do not have ground truth depth, we evaluate the difference between the depth decoded by our depth head and the depth labels predicted by VideoDepthAnything on the DL3DV dataset, as VideoDepthAnything predicted labels are used as training supervision in our method.
>
>
>     | Heads           | MSE   |
>     | --------------- | ----- |
>     | Our depth head | 0.0014 |
>     | Video VAE Decoder | 0.0011 |
>
> 4. **Writing errors.**
>
>     Regarding the writing issues, we will carefully revise the manuscript from beginning to end to correct such errors. We sincerely thank you for your careful attention and feedback.
>
> [1] Duan H, Yu H X, Chen S, et al. Worldscore: A unified evaluation benchmark for world generation[J]. arXiv preprint arXiv:2504.00983, 2025.

---

### Decision · Program_Chairs · 2025-09-17

**Decision:**

Accept (poster)

**Comment:**

This work makes a valuable contribution to video generation by demonstrating that explicit geometric supervision can significantly improve temporal consistency without sacrificing visual quality. The technical approach is sound, the experimental validation is comprehensive, and the practical benefits are clear. The consistent positive reviewer feedback and successful resolution of all concerns through detailed rebuttals support acceptance. The plug-and-play nature and demonstrated improvements across multiple metrics make this a valuable addition to the video generation toolkit, with broad applicability to existing frameworks.